# Effects of Branched-Chain Amino Acid (BCAA) Supplementation on the Progression of Advanced Liver Disease: A Korean Nationwide, Multicenter, Prospective, Observational, Cohort Study

**DOI:** 10.3390/nu12051429

**Published:** 2020-05-15

**Authors:** Jung Gil Park, Won Young Tak, Soo Young Park, Young Oh Kweon, Woo Jin Chung, Byoung Kuk Jang, Si Hyun Bae, Heon Ju Lee, Jae Young Jang, Ki Tae Suk, Myung Jin Oh, Jeong Heo, Hyun Young Woo, Se Young Jang, Yu Rim Lee, June Sung Lee, Do Young Kim, Seok Hyun Kim, Jeong Ill Suh, In Hee Kim, Min Kyu Kang, Won Kee Lee

**Affiliations:** 1Department of Internal Medicine, College of Medicine, Yeungnam University, Daegu 42415, Korea; gsnrs@naver.com (J.G.P.); hjlee@med.yu.ac.kr (H.J.L.); kmggood111@naver.com (M.K.K.); 2Department of Internal Medicine, School of Medicine, Kyungpook National University, Kyungpook National University Hospital, Daegu 41944, Korea; psyoung0419@gmail.com (S.Y.P.); yokweon@knu.ac.kr (Y.O.K.); magnolia1103@naver.com (S.Y.J.); deblue00@naver.com (Y.R.L.); 3Department of Internal Medicine, School of Medicine, Keimyung University, Daegu 42601, Korea; chung50@dsmc.or.kr (W.J.C.); jangha106@dsmc.or.kr (B.K.J.); 4Department of Internal Medicine, College of Medicine, The Catholic University of Korea, Seoul 06591, Korea; baesh@catholic.ac.kr; 5Department of Internal Medicine, College of Medicine, Soonchunhyang University, Seoul 04401, Korea; jyjang@hosp.sch.ac.kr; 6Department of Internal Medicine, College of Medicine, Hallym University, Chuncheon 24252, Korea; ktsuk@hallym.ac.kr; 7Department of Internal Medicine, CHA Gumi Medical Center, CHA University School of Medicine, Gumi 39295, Korea; zenus1@hanmail.net; 8Department of Internal Medicine, School of Medicine, Pusan National University, Pusan 49241, Korea; jheo@pusan.ac.kr (J.H.); who54@hanmail.net (H.Y.W.); 9Department of Internal Medicine, Ilsan Paik Hospital, College of Medicine, Inje University College of Medicine, Goyang 10380, Korea; jsleemd@paik.ac.kr; 10Department of Internal Medicine, College of Medicine, Yonsei University, Seoul 03722, Korea; dyk1025@yuhs.ac; 11Department of Internal Medicine, School of Medicine, Choungnam National University, Daejeon 61469, Korea; midoctor@cnu.ac.kr; 12Department of Internal Medicine, College of Medicine, Dongguk University, Gyeongju 39067, Korea; sujungil@dongguk.ac.kr; 13Department of Internal Medicine, School of Medicine, Chonbuk National University, Chungju 54907, Korea; ihkimmd@chonbuk.ac.kr; 14Medical Research Collabration Center in KNUH and School of Medicine, Kyungpook National University, Daegu 41944, Korea; wonlee@knu.ac.kr

**Keywords:** amino acids, branched-chain, liver cirrhosis, prognosis, ascites, hepatic encephalopathy

## Abstract

Background and Aims: Clinical evidence for the benefits of branched-chain amino acids (BCAAs) is lacking in advanced liver disease. We evaluated the potential benefits of long-term oral BCAA supplementation in patients with advanced liver disease. Methods: Liver cirrhosis patients with Child–Pugh (CP) scores from 8 to 10 were prospectively recruited from 13 medical centers. Patients supplemented with 12.45 g of daily BCAA granules over 6 months, and patients consuming a regular diet were assigned to the BCAA and control groups, respectively. The effects of BCAA supplementation were evaluated using the model for end-stage liver disease (MELD) score, CP score, serum albumin, serum bilirubin, incidence of cirrhosis-related events, and event-free survival for 24 months. Results: A total of 124 patients was analyzed: 63 in the BCAA group and 61 in the control group. The MELD score (*p* = 0.009) and CP score (*p* = 0.011) significantly improved in the BCAA group compared to the control group over time. However, the levels of serum albumin and bilirubin in the BCAA group did not improve during the study period. The cumulative event-free survival was significantly improved in the BCAA group compared to the control group (HR = 0.389, 95% CI = 0.221–0.684, *p* < 0.001). Conclusion: Long-term supplementation with oral BCAAs can potentially improve liver function and reduce major complications of cirrhosis in patients with advanced liver disease.

## 1. Introduction

Protein-calorie malnutrition is a progressing condition caused by underlying chronic liver disease and poor nutritional support [1,2]. As chronic liver disease progresses to decompensated cirrhosis, more than 60% of patients suffer from malnutrition, which worsens their quality of life and increases their mortality rate [1,3]. Although most of the conditions underlying chronic liver disease, including chronic hepatitis B (CHB) and chronic hepatitis C, have been relatively well controlled by recent advances in antiviral treatments, malnutrition is frequently underevaluated in clinical practice [4]. The adverse impacts of malnutrition are related to the progression of liver disease in patients with alcoholic liver disease [5]. Sarcopenia, which is characterized by a low skeletal muscle mass, is also known to be related to the prognosis in patients with any etiology of chronic liver disease, including nonalcoholic fatty liver disease [6,7].

As hepatic function worsens, decreasing hepatic ureagenesis impairs the disposal pathway for blood ammonia. In addition, impairment of plasma protein synthesis induces hypoproteinemia and decreases the skeletal muscle mass [8]. Initially, branched-chain amino acids (BCAAs) are not catabolized in the liver but in peripheral tissues due to lack of BCAA aminotransferase (BCAT) in the liver [9]. Though skeletal muscle has high BCAT activity, which is a major organ for BCAA catabolism, the liver has the highest activity of branched-chain α-keto acid dehydrogenase for the cooperation of BCAA catabolism [10]. The demand for BCAAs is increased in cirrhotic patients with portosystemic shunt and malnutrition owing to the ammonia being converted into glutamine in skeletal muscle and brain [4]. The depletion of BCAAs is aggravated by the hypermetabolic state of cirrhosis, poor oral intake, and malabsorption [11].

The beneficial effects of BCAA supplementation in liver disease include improvements in body composition and nitrogen balance, liver cell regeneration, protein and albumin synthesis, hepatic encephalopathy, and immune function [12]. However, a recently updated Cochrane review concluded that there were no beneficial effects on mortality, quality of life, or parameters related to nutrition other than hepatic encephalopathy [13]. Unfortunately, most of the studies analyzed in that review were too small to provide evidence of satisfactory quality, and so additional studies are needed to delineate the beneficial effects of this nutritional intervention, including hypoalbuminemia; increased muscle mass; and cirrhosis-related complications such as infection, gastrointestinal bleeding, ascites, and hepatic encephalopathy [14].

The present study evaluated the long-term beneficial effects of oral BCAA supplementation by analyzing various parameters of hepatic function, including the model for end-stage liver disease (MELD) and Child–Pugh (CP) score, as well as the development of cirrhosis-related events and hepatocellular carcinoma (HCC).

## 2. Materials and Methods

### 2.1. Study Population

From January 2013 to June 2017, we screened 232 consecutive patients who were clinically or pathologically diagnosed as having advanced liver cirrhosis with CP scores from 8 to 10 at 14 medical centers in the Republic of Korea. The main exclusion criteria were viable HCC, advanced Barcelona Clinic Livre Cancer stage, HCC with a life expectancy of less than 6 months, a recent diagnosis of other malignancy (within 3 years), finding it impossible to cease alcohol consumption during study period, serum creatinine level higher than 1.5 mg/dL at the time of enrollment, early liver transplantation within 6 months after enrollment, and metabolic disorders presenting branched-chain ketoaciduria; these criteria were also used in a previous retrospective study conducted by our group [15].

### 2.2. Study Design

The enrolled patients were assigned to two groups: no treatment (control) or treatment with BCAAs. The BCAA group was treated with 12.45 g of BCAA granules (LIVACT, Samil Pharmaceutical, Seoul, Korea; 4.15 g of BCAA granules per sachet containing 952 mg of L-isoleucine, 1904 mg of L-leucine, and 1144 mg of L-valine) for at least 6 months (Figure 1).

The adherence of patients in the BCAA group was considered to be adequate, at more than 80%, which was assessed by each investigator in clinical setting. The control group consumed a standard diet without BCAAs or with only a short course of BCAA treatment lasting less than 1 month. Patients who had previously consumed BCAAs for more than 1 month were excluded at the baseline screening from both groups. There was no further dietary intervention of patients except standard dietary counseling of cirrhosis in both groups [14].

The primary end point of this study was the change in MELD score over time. The secondary end points included the changes in the CP score and serum albumin, and the development and recurrence of HCC, mortality, and cirrhosis-related complications; these end points were also used in the previous retrospective study conducted by our group [15].

Written informed consent was obtained from all patients in both groups. This study was approved by the institutional review board of each center, and it was conducted in accordance with the principles of the Declaration of Helsinki.

### 2.3. Baseline Clinical and Laboratory Assessments

The medical history was initially assessed, including previously diagnosed malignancies, medications that could affect prothrombin activity (e.g., warfarin), recent regular albumin replacement, previous and current alcohol consumption, previous and current antiviral treatment for hepatitis B virus (HBV) or hepatitis C virus (HCV), and previous cirrhosis-related complications. The initial laboratory data included liver function tests, serum creatinine, viral markers for HBV and HCV, titers of HBV DNA and HCV RNA (if the patients had HBsAg or HCV antibodies), prothrombin time, CP score, and MELD score. CP score was calculated by each investigator using assigned points of serum total bilirubin, serum albumin, prothrombin time, and grades of ascites and hepatic encephalopathy at the time of enrollment [16]. MELD score was calculated as 9.57 × log (creatinine, mg/dL) + 3.78 log (bilirubin, mg/dL) + 11.2 × log (INR) + 6.43 (constant for liver disease etiology). Any values used to calculate MELD score that were less than one were converted to 1.0 [17]. All of the laboratory data were assessed at least 3 months after resolution of acute liver-related events, including variceal bleeding, acute infection, hepatic encephalopathy according to West Haven criteria [18], hepatorenal syndrome, and other acute medical conditions that could affect the CP or MELD score.

### 2.4. Follow-up Clinical and Laboratory Assessments

To avoid practice variation of all centers, all surveillance of variceal and HCC, and prophylaxis of varices and spontaneous bacterial peritonitis, adhered to the following current guidelines [14]. Cirrhosis-related complications were assessed by current guideline as well [14]. The event of cirrhosis-related complication was defined as any medical conditions related to liver cirrhosis that needed intervention or admission. Major cirrhosis-related events were defined as rupture of varices, development or aggravation of ascites, hepatorenal syndrome, hepatic encephalopathy, spontaneous bacterial peritonitis, development or recurrence of HCC, death from any cause, and any life-threatening medical conditions related to liver cirrhosis except malignancy. The first event of each complication was counted for complications to compare the development of each complication in both groups. Regardless of its kind, only the first event was counted for cirrhosis-related events to analyze cumulative event-free survival. Follow-up data were obtained after 6 months of BCAA treatment or consuming a regular diet from the BCAA and control groups, respectively. The clinical data included the reason for visiting the clinic, consumption of alcohol, development and recurrence of HCC, death from any cause, and liver-related complications. The laboratory data included liver function tests, serum creatinine, prothrombin time, CP score, and MELD score, which were assessed at the time of visiting the clinic.

### 2.5. Statistical Analysis

Data were analyzed using IBM SPSS software (version 23.0, IBM, Armonk, NY, USA). Baseline characteristics were compared between the two study groups using the chi-square test, Student’s *t* test, Wilcoxon rank sum test, or a linear-by-linear association test. The changes in the MELD score, CP score, serum bilirubin, and albumin between the two groups were analyzed using a mixed linear model. We compared the incidence of liver-related complications, development and recurrence of HCC, and death using the chi-square test or Fisher’s exact test. The cumulative event-free survival (EFS) rates were analyzed using the Kaplan–Meier method, and compared using the log-rank test. We counted the number of patients lost to follow-up or with cirrhosis-related complications or death from any cause in the analysis. The development and recurrence of HCC were analyzed using Fisher’s exact test, while factors associated with HCC could not be analyzed due to the low incidence of HCC. A probability value of *p* < 0.05 was considered statistically significant.

## 3. Results

This study screened 232 patients for eligibility, which resulted in the exclusion of 5 patients with viable HCC, other untreated malignancy, or serum creatinine above 1.5 mg/dL. During the 6-month window period for data inclusion, 104 patients dropped out for the following reasons: follow-up loss (*n* = 61), inadequate consumption of BCAAs (*n* = 19), lack of follow-up data (*n* = 14), alcohol consumption (men > 30g/day; women > 20g/day, *n* = 6), resolution of acute liver-related events (*n* = 3), and early liver transplantation (*n* = 1). Finally, 124 patients (63 in the BCAA group and 61 in the control group) were followed up for additional 18 months and analyzed (Figure 2).

### 3.1. Baseline Characteristics of the Patients

The baseline characteristics did not differ significantly between the two groups (Table 1).

The median follow-up duration also did not differ between the two groups, being 15.2 months (range = 8.0–19.3 months) in the control group and 16.6 months (range = 11.0–22.2 months) in the BCAA group (*p* = 0.111). The median duration of BCAA consumption in the BCAA group was 20.1 months (range = 11.0–24.0 months). Among the patients with CHB, antiviral agents, including tenofovir and entecavir, were started due to a high viral load (above 2000 IU/mL) at the time of study enrollment in seven patients in the BCAA group and five patients in the control group. The serum HBV DNA levels of the others were below 116 copies/mL regardless of antiviral treatment. Among the patients with hepatitis C antibodies, HCV RNA was detected in only two patients in the BCAA group, and they were not treated with interferon plus ribavirin, or direct-acting agents due to the presence of decompensated cirrhosis or drug nonavailability. 

### 3.2. Outcomes Related to Liver Function

The changes in the MELD score, CP score, serum albumin, and bilirubin over 2 years are compared between the two groups in Figure 3. The MELD and CP scores improved significantly in the BCAA group over time compared to the control group (*p* = 0.009 and *p* = 0.011, respectively). However, the improvements in serum albumin and bilirubin did not differ significantly over time between the two groups (*p* = 0.149 and *p* = 0.233, respectively). In the subgroup analysis, an improvement in serum albumin was not demonstrated in patients with serum albumin at 3.5 mg/dL or less, with improvement only observed in those consuming BCAAs (*p* = 0.046).

### 3.3. Cirrhosis-Related Events and HCC

The incidence rates of major cirrhosis-related events and HCC are presented in Table 2. Major cirrhosis-related events occurred less in the BCAA group than in the control group (*p* < 0.001), among which the development or aggravation of ascites and hepatic encephalopathy occurred less in the BCAA group (*p* = 0.017 and *p* = 0.046, respectively). However, the mortality rate did not differ significantly between the two groups. The cumulative EFS was significantly better in the BCAA group than in the control group: 605 ± 26 days (95% CI = 555 – 655 days) vs. 486±28 days (95% CI = 432–540 days) (*p* < 0.001, Figure 4). The development and recurrence of HCC did not differ between the two groups (*p* = 0.678 and *p* = 1.000, respectively).

## 4. Discussion

This study evaluated the beneficial effects of BCAAs in cirrhotic patients with CP scores from 8 to 10 who were in the decompensated stage rather than the terminal stage. The patients supplemented with BCAAs for more than 6 months exhibited improved MELD and CP scores compared to those did not consume BCAAs. Even if treating the underlying liver disease is the most important intervention, nutritional support is also known to be an independent factor for improving the outcomes in these patients [4]. Since the underlying liver disease could induce disease progression, all of the patients in the present study were treated appropriately according to the etiology of their liver disease. All of the included patients abstained from alcohol consumption and were treated with antiviral agents according to guidelines when they had chronic viral hepatitis. Thus, most of the clinical parameters, including MELD score, CP score, serum albumin, and serum bilirubin, were improved in both groups (Figure 2). However, additional beneficial effects on the MELD and CP scores were demonstrated in the patients supplemented with BCAAs. An Italian randomized prospective study also found improvements in CP scores after supplementation with BCAAs in advanced liver disease [2]. However, there have been no reports based on prospective studies of improvements in MELD scores after supplementation with BCAAs.

Major cirrhosis-related events occurred less frequently in the BCAA group than in the control group. Our previous retrospective study failed to show improvement of EFS [15]. The most common etiology in that previous study was chronic hepatitis B, while it was alcoholic liver disease in the present study. Unlike other etiologies of liver cirrhosis, ethanol and its metabolites exert direct effects on the worsening of sarcopenia. Both of these compounds reduce protein synthesis and increase autophagy via the activation of critical regulatory downstream pathways involving myostatin or dephosphorylation of mTORC1 and DRP1 [19]. It is therefore possible that these differences in the etiologies of the analyzed patients are due to the conflicting data.

Among major cirrhosis-related events, the development or aggravation of ascites and hepatic encephalopathy occurred less frequently in the BCAA group than in the control group. Moreover, the serum albumin level did not improve in the BCAA group throughout the study period. However, among the patients with a low albumin level (≤3.5 mg/dL), the serum albumin level improved in the BCAA group compared to the control group. A Japanese prospective study found improvement of serum albumin in cirrhotic patients with serum albumin at 3.5 g/dL or less and the presence of ascites, peripheral edema, or hepatic encephalopathy [20]. The findings of our subgroup analysis are also supported by a recent study demonstrating that BCAA supplementation improved the survival rate in cirrhotic patients with sarcopenia and a low albumin level (≤3.5 mg/dL) [21]. It is therefore possible that this subgroup contributed to the improvement of ascites control in the BCAA group.

While no improvement in hepatic encephalopathy was found in our previous retrospective study, the present study found that hepatic encephalopathy developed less frequently in the BCAA group [15]. It is possible that the early withdrawal of BCAA supplementation in the previous study is responsible for these conflicting data. Although the beneficial effect of BCAAs has been shown in several studies, the efficacy of oral BCAAs seems to be modest, and BCAAs are relatively expensive [22]. Thus, oral BCAAs would be considered as an alternative option in patients with hepatic encephalopathy.

The development and recurrence rate of HCC was also analyzed in the present study, but the relatively small number of patients could have been responsible for no conclusive evidence for the anticarcinogenic effect of BCAAs being found. An in vitro study produced evidence that the anticarcinogenic effects of BCAAs are related to down-regulation of insulin-like growth factor-1 receptor and vascular endothelial growth factor [23,24]. However, its efficacy in clinical studies has only been shown in relatively small numbers of participants or limited subgroups, such as obese men with high alpha-fetoprotein [25,26].

The present study was subject to some limitations. First, this study did not have a randomized controlled design, and so although it was conducted prospectively, selection bias could have influenced the results. Second, the indication and duration of BCAA supplementation varied between the included centers, and the dietary habits of the subjects were not tracked. However, BCAA supplementation is required for at least 6 months, and approximately 70% of the patients in the BCAA group completed the course of BCAA supplementation during the follow-up period. Third, we could not evaluate the nutritional status (including sarcopenia) based on anthropometric parameters such as the midarm muscle circumference, or functional performance such as gait speed or handgrip strength in this study. However, in the subgroup analysis the level of serum albumin—indicating the nutritional and hepatic functional status—increased in patients with a low albumin level during BCAA supplementation. To maintain a better serum albumin level, long-term supplementation with oral BCAAs should be considered. In addition, future studies should evaluate the beneficial effects of BCAAs in cirrhotic patients with sarcopenia.

In conclusion, this study has demonstrated that long-term supplementation with oral BCAAs improved hepatic reservoir parameters, including the MELD and CP scores in patients with advanced liver disease. The patients supplemented with oral BCAAs exhibited fewer cirrhosis-related complications, especially the development or aggravation of ascites and hepatic encephalopathy. Thus, long-term supplementation with oral BCAAs could improve the clinical outcomes of patients with advanced liver disease.

## Figures and Tables

**Figure 1 nutrients-12-01429-f001:**
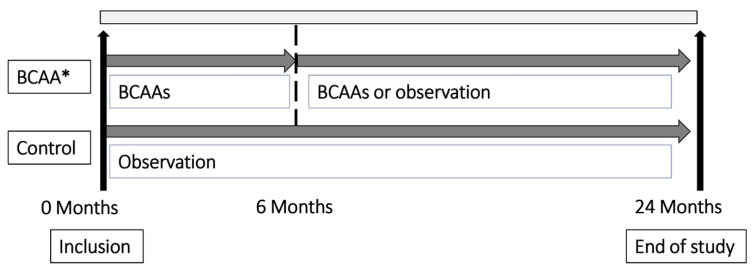
Scheme of the study. * Adherence of patients to prescribed BCAAs was > 80%. BCAA, branched-chain amino acid.

**Figure 2 nutrients-12-01429-f002:**
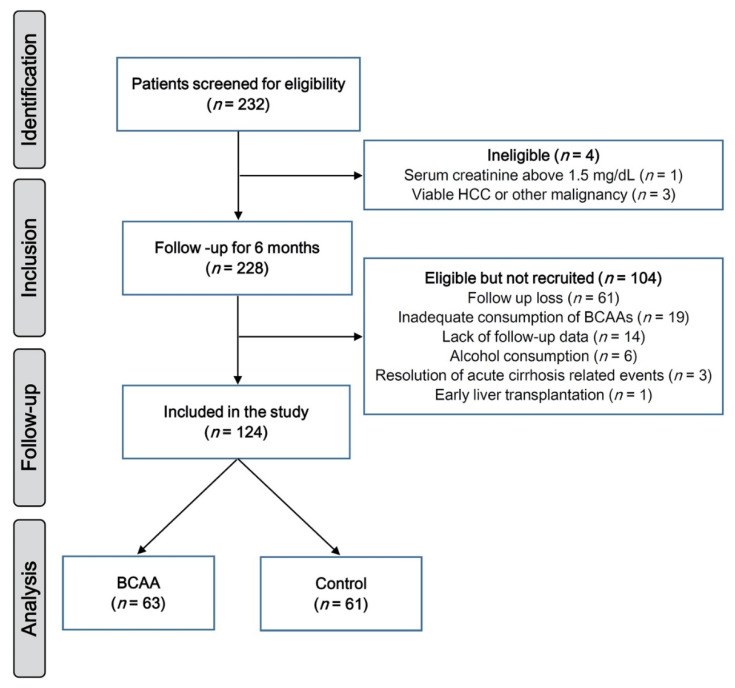
Flow diagram of the study. HCC, hepatocellular carcinoma; BCAA, branched-chain amino acid.

**Figure 3 nutrients-12-01429-f003:**
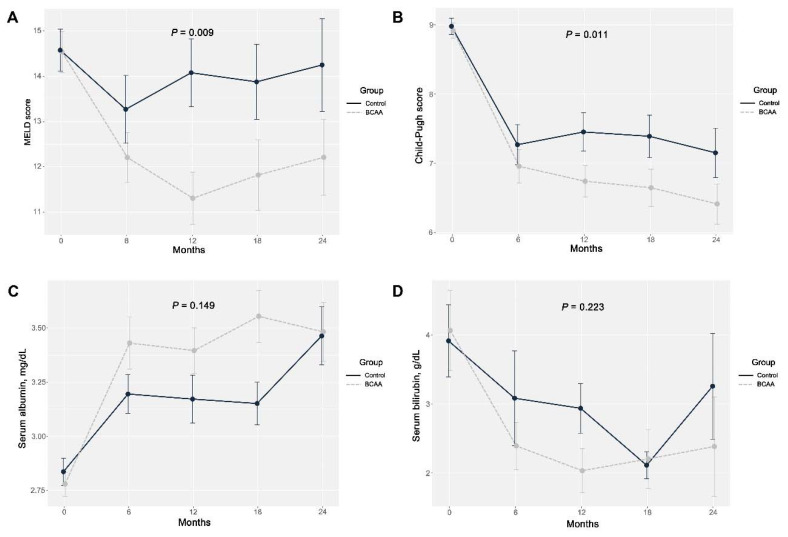
Changes in the model for end-stage liver disease score (**A**), Child–Pugh score (**B**), serum albumin (**C**), and total bilirubin (**D**) in the BCAA and control groups over 2 years. MELD, model for end-stage liver disease; BCAA, branched-chain amino acid.

**Figure 4 nutrients-12-01429-f004:**
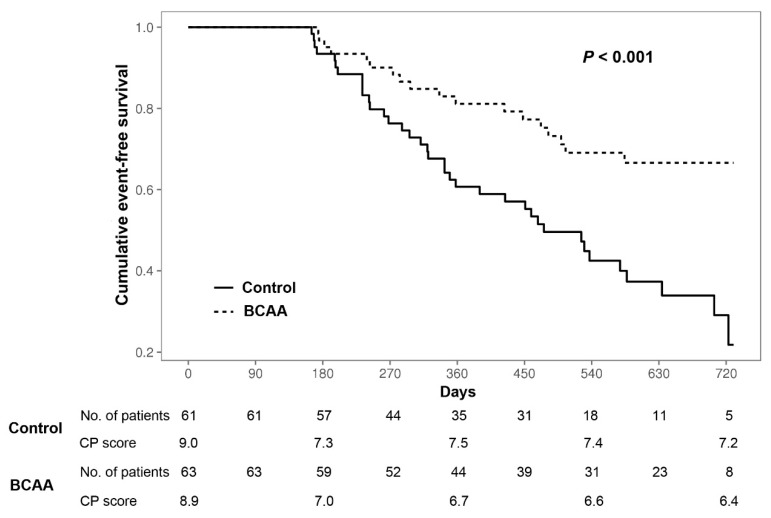
Cumulative cirrhosis-related event-free survival in the BCAA and control groups. BCAA, branched-chain amino acid; CP, Child–Pugh.

**Table 1 nutrients-12-01429-t001:** Baseline characteristics in the branched-chain amino acid (BCAA) and control groups.

Characteristic	BCAA	Control	*p* Values
Number of patients	63	61	-
Sex, male	45 (71)	37 (61)	0.281
Age, years	60 ± 10	58 ± 11	0.742
BMI, kg/m^2^	23.0 [21.3-25.1]	22.2 [20.2-24.2]	0.136
Etiology HBV/HCV/alcohol/other	17/6/31/9 (27/10/49/14)	15/3/38/5 (25/5/62/8)	0.400
Child–Pugh score 8/9/10	27/15/21 (43/24/33)	23/16/22 (38/26/36)	0.842
MELD score	14.5 [12.1-16.8]	14.2 [11.9-16.1]	0.914
AST, IU/L	62.0 [36.0-106.0]	59.0 [37.0-134.0]	0.772
ALT, IU/L	31.0 [20.0-41.0]	31.0 [18.0-57-5]	0.447
γ-glutamyl transferase, IU/L	73.0 [32.0-175.0]	106 [25.0-315.0]	0.604
Serum albumin, g/dL	2.7 [2.5-3.1]	2.8 [2.5-3.1]	0.692
Total bilirubin, mg/dL	2.5 [1.9-4.1]	2.7 [2.0-3.8]	0.974
Blood urea nitrogen, mg/dL	11.8 [9.0-16.9]	11.9 [9.2-17.0]	0.675
Serum creatinine (mg/dL)	0.8±0.2	0.8±0.2	0.711
INR	1.4 [1.3-1.5]	1.4 [1.2-1.5]	0.649
Platelet count, ×10^9^/L	100.0 [64.0-134.5]	81.0 [57.0-103.0]	0.079
History of variceal bleeding	47 (74.6)	39 (63.9)	0.274
Hepatic encephalopathy none/grade 1 or 2/grade 3 or 4	57/6/0 (90/10/0)	53/7/1 (87/11/2)	0.551
Ascites none/mild/moderate to severe	11/38/14 (18/60/22)	8/39/14 (13/64/23)	0.797
History of HCC	11 (18)	8 (13)	0.673

Data are mean ± SD, median [range] or *n* (%) values. BMI, body mass index; HBV, hepatitis B virus; HCV, hepatitis C virus; MELD, model for end-stage liver disease; AST, aspartate aminotransferase; ALT, alanine aminotransferase INR, international normalized ratio; HCC, hepatocellular carcinoma.

**Table 2 nutrients-12-01429-t002:** Major cirrhosis-related events in the BCAA and control groups.

Event	BCAA	Control	*p* Values
Number of patients	63	61	-
Total events except HCC	14 (29)	37 (61)	0.001
Rupture of varices	2 (3)	6 (10)	0.253
Development or aggravation of ascites	6 (10)	17 (28)	0.017
Hepatorenal syndrome	1 (2)	1 (2)	1.000
Hepatic encephalopathy	6 (10)	15 (25)	0.046
Spontaneous bacterial peritonitis	2 (3)	1 (2)	1.000
Other *	0 (0)	1 (2)	0.987
Development of HCC	14 (22)	10 (16)	0.553
Recurrence of HCC	7 (11)	4 (7)	0.565
Death	6 (10)	9 (15)	0.537

Data are *n* (%) values. BCAA, branched-chain amino acid; HCC, hepatocellular carcinoma. * Other = retroperitoneal hemorrhage.

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
