# Peer review of "Effects of Branched-Chain Amino Acid (BCAA) Supplementation on the Progression of Advanced Liver Disease: A Korean Nationwide, Multicenter, Prospective, Observational, Cohort Study"

_nutrients, 2020, doi:10.3390/nu12051429_

Round 1
Reviewer 1 Report
This is a very interesting study apparently dispelling the paradox of protein over-load as BCAA leads to increase in hepatic encephalopathy. There are several points that you need to address in this study. It would be of value to add some additional data and revise the manuscript as detailed below. This is a clinically important study that needs to be disseminated the clinician and basic scientist.
You need to supply definition of all the abbreviations.
You also need to report how MELD and CP scores are calculated since many readers of this manuscript are not clinicians
Line 67: BCAA are degraded both in the muscle and brain. Which other human tissues, kidney, lung, etc. have the BCAA dehydrogenase and aminotransferase should be reviewed and reported in this study since it may give insight in the beneficial effects of BCAA in patients with advanced liver disease.
The use of BCAA by the brain is of particular importance in hepatic encephalopathy (HE) since they contribute to the TCA pool and functions as a cataplerotic avenue for glutamate production in use to capture ammonia thus leading to an increase in glutamine. Therefore, it of importance that you report the neurological score of your patients (West-Haven or Glasgow score) to determine if BCAA reduce ammonia and HE.
Have you looked at altering the dose of BCAA and whether the inclusion of zinc with BCAA may improve outcomes. You need to discuss the present-day treatments and why and how they improve outcomes in patients with advanced liver disease.
You need to provide a possible mechanism by which BCAA supplementation improves liver cirrhosis.
Please include a table of clinical values of BCAA and control groups with those values identifying liver function (prothrombin time, bilirubin, etc.) and liver damage (AST, ALT, AP, etc.). It is of extreme importance to provide values on serum ammonia levels, blood urea nitrogen, and total serum protein. It would also be of value to report serum acute phase response protein levels (Hemopexin, haptoglobin, etc.)
What is the reason for the plateauing of serum albumin in the BCAA group compared to the control group?
Author Response
Thank you for good comments.
We response your review as below.
And, revised contents was written in red color on revised manuscript.
You need to supply definition of all the abbreviations.
All the abbreviations are added as your comment.
You also need to report how MELD and CP scores are calculated since many readers of this manuscript are not clinicians
On page 7, line 4, we added description of value for calculation of CP and MELD score briefly for non-clinician or non-hepatologist reader.
Line 67: BCAA are degraded both in the muscle and brain. Which other human tissues, kidney, lung, etc. have the BCAA dehydrogenase and aminotransferase should be reviewed and reported in this study since it may give insight in the beneficial effects of BCAA in patients with advanced liver disease.
Thank your good comment. We added review of BCAA aminotransferase and branched-chain α-keto acid dehydrogenase to give insight in the beneficial effects of BCAA in patients with liver cirrhosis and muscle wasting in 14th – 18th line of introduction.
The use of BCAA by the brain is of particular importance in hepatic encephalopathy (HE) since they contribute to the TCA pool and functions as a cataplerotic avenue for glutamate production in use to capture ammonia thus leading to an increase in glutamine. Therefore, it of importance that you report the neurological score of your patients (West-Haven or Glasgow score) to determine if BCAA reduce ammonia and HE.
As our study is prospective observational study, it is not allowed to assess West-Haven or Glasgow score in all patients including those without HE. If our study includes these neurological score, ethical committee would not permit our study protocol due to violation of good clinical practice.
Have you looked at altering the dose of BCAA and whether the inclusion of zinc with BCAA may improve outcomes. You need to discuss the present-day treatments and why and how they improve outcomes in patients with advanced liver disease.
Thank you for good comment. micronutrients such as zinc supplementation with BCAA could improve nitrogen balance, tracking of dietary habits is important to delineate efficacy of BCAA supplementation in patients with liver cirrhosis (Hepatology research : the official journal of the Japan Society of Hepatology 2007, 37, 615-619, doi:10.1111/j.1872-034X.2007.00095.x). At the time of setting a study design, we did not consider micronutrient intervention or analysis, because we could not measure these in clinical practice as above reason. In addition, it is not focus on our study.
You need to provide a possible mechanism by which BCAA supplementation improves liver cirrhosis.
There are lots of mechanism of BCAA supplementation improves liver cirrhosis including ammonia detoxification to glutamine in muscles, liver regeneration, albumin synthesis, immune and hepatic function, glucose metabolism, and physical and metal fatigue, which was already described on line 1st – 3rd of page 6 (introduction). However, its results from clinical evidence is weak. We think different indication of BCAA supplementation in real practice cause this discrepancy. The clinical data provided in this manuscript is the strength of our study. If you think it is not enough for reader, we would provide additional description.
Please include a table of clinical values of BCAA and control groups with those values identifying liver function (prothrombin time, bilirubin, etc.) and liver damage (AST, ALT, AP, etc.). It is of extreme importance to provide values on serum ammonia levels, blood urea nitrogen, and total serum protein. It would also be of value to report serum acute phase response protein levels (Hemopexin, haptoglobin, etc.)
On table 1, INR (PT), bilirubin identifying liver function is already shown. As your comment, we added AST, ALT, GGT, BUN on table 1. However, we don’t have data for AP, ammonia, serum protein, and acute phase response protein. We could not include all of other laboratory test, because enrolled patients were between CP 7-9 with or without HE. It is same reason as above.
In addition, we have revised value of laboratory test without normal distribution in table 1 with Wilcoxon rank sum test for better analysis.
What is the reason for the plateauing of serum albumin in the BCAA group compared to the control group?
Thank you for good question. As our results, improvement of albumin is shown in patients with a low albumin (≤3.5 mg/dL). I think that is the reason why the plateauing of serum albumin is shown near the level of 3.5 mg/dL. Although improvement of serum albumin is shown in both control and BCAA group, it is more dramatically increased in BCAA group. A Japanese prospective study also supports that finding. (Clinical gastroenterology and hepatology. 2005, 3, 705-713) In addition, the patients enrolled in our study had decompensated but not far advanced liver cirrhosis, which were hard to recover completely without liver transplantation.

Reviewer 2 Report
The authors show effect of BCAA on early liver cirrhosis in Korean population. The effect of BCAA in cirrhosis is well studies, hence the novelty of the study is in Korean population. The treatment with BCAA definitely improved survival which is the strongest point.The recruitment and screening done in study is solid. However the outcome is not significant enough to back the claims made.
- Figure 1 suggest the Child-Pugh score is significantly different, however the serum albumin and serum bilirubin is not. Additionally, the encephalopathy and ascites events are also similar. Authors need to comment or show if scores for ascites and encephalopahty were high. Since they comprise the Child-Pugh score.
- Table 2 show no difference in any of the criteria when compared in total percent of events. Which weakens the study results.
Author Response
Thank you for good comments.
We response your review as below.
And, revised contents was written in red color on revised manuscript.
Open Review 2
1. Figure 1 suggest the Child-Pugh score is significantly different, however the serum albumin and serum bilirubin is not. Additionally, the encephalopathy and ascites events are also similar. Authors need to comment or show if scores for ascites and encephalopahty were high. Since they comprise the Child-Pugh score.
Thank you for good comments. I think you mean Figure 3. As your comments, development or aggravation of ascites and HE can be shown in same time. Therefore, event free survival is more important to delineate efficacy of BCAA supplementation, which is one of the important secondary end points of this study. At the same time, CP score is good parameter to reflect hepatic reservoir but assigned point of ascites and HE are relatively subjective. Therefore, our primary end point of this study is MELD score, which is objective parameter of hepatic reservoir. Though we did not show assigned point of ascites and HE, we provided changes of serum albumin and bilirubin, which is two of five factors consisted of CP score over time in figure 3.
2. Table 2 show no difference in any of the criteria when compared in total percent of events. Which weakens the study results.
In table 2, total event except HCC is 29% in BCAA group and 61% in control group (P = 0.001). Especially, ascites and HE is more frequently developed in control group. But the number of other events including rupture of varcies, HRS, SBP and death is too small to show statistical difference when they were separately analyzed.

Round 2
Reviewer 1 Report
none
Author Response
Thank you for kind review.

Reviewer 2 Report
Thanks for the clarification however authors need to show the breakdown of scoring atleast in tabular or graphical manner. This will help understand the readers the impact of BCAA better.
For Table 2, Authors need to clarify how the percentage was calculated, what was counted for total percentage. Since most of the events on individual comparison doesnt show much changes. This observation is crucial as it will support the improved survival data showed in this paper.
Author Response
Comments and Suggestions for Authors
Thanks for the clarification however authors need to show the breakdown of scoring atleast in tabular or graphical manner. This will help understand the readers the impact of BCAA better.
Thank you for good comment. As your comment, we added figure 4, which shows percentage of major cirrhosis-related events in the BCAA and control group with graphical manner to help readers understand.
Figure 4. Major cirrhosis-related events in the BCAA and control groups
For Table 2, Authors need to clarify how the percentage was calculated, what was counted for total percentage. Since most of the events on individual comparison doesnt show much changes. This observation is crucial as it will support the improved survival data showed in this paper.
As your comment, we added descriptions of table 2 section of MATERIALS AND METHOD (2.4 Follow-up clinical and laboratory assessments) to support event-free survival data as below
‘The first event of each complication was counted for complication to compare the development of each complication in both group. Regardless of its kinds, only the first event was counted for cirrhosis-related events to analyze cumulative event-free survival.’
